# TRELLISWorld: Training-Free World Generation from Object Generators

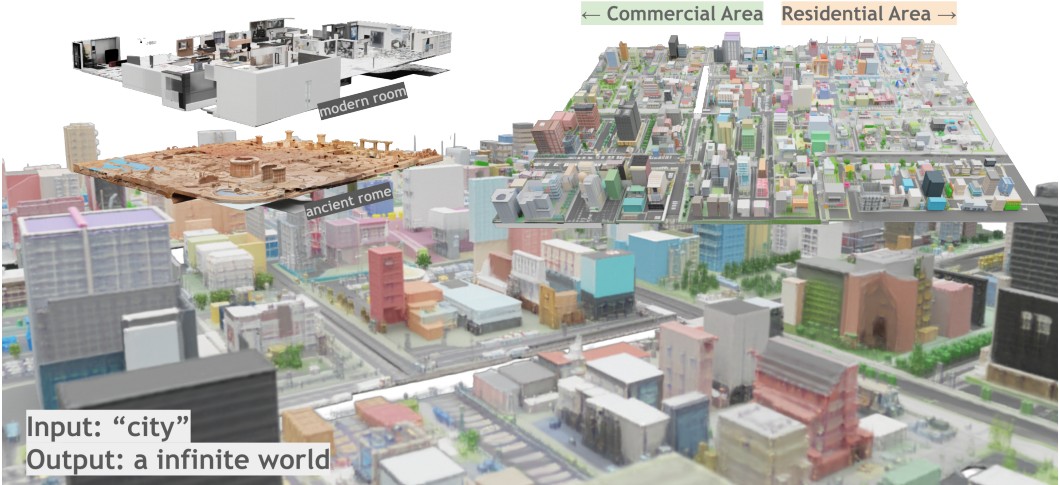

Figure 1: Scenes generated by our framework, **TRELLISWorld**, using only natural language input. Users may provide fine-grained prompts for specific regions, enabling semantically consistent gradual transitions: e.g., from a dense commercial district with greens, into low-density residential zones.

## Abstract

Text-driven 3D scene generation holds promise for a wide range of applications, from virtual prototyping to AR/VR and simulation. However, existing methods are often constrained to single-object generation, require domain-specific training, or lack support for full 360-degree viewability. In this work, we present a training-free approach to 3D scene synthesis by repurposing general-purpose text-to-3D object diffusion models as modular tile generators. We reformulate scene generation as a multi-tile denoising problem, where overlapping 3D regions are independently generated and seamlessly blended via weighted averaging. This enables scalable synthesis of large, coherent scenes while preserving local semantic control. Our method eliminates the need for scene-level datasets or retraining, relies on minimal heuristics, and inherits the generalization capabilities of object-level priors. We demonstrate that our approach supports diverse scene layouts, efficient generation, and flexible editing, establishing a simple yet powerful foundation for general-purpose, language-driven 3D scene construction. We will release the full implementation upon publication.

## 1 Introduction

Generating 3D worlds from text represents a longstanding goal at the intersection of computer graphics, machine learning, and human-computer interaction. Enabling users to describe a virtual world in natural language and synthesize an editable 3D environment would transform multiple domains. For example, in creative content design, this could accelerate ideation

workflows, allowing a game designer to rapidly prototype a level using coarse spatial prompts, or enabling a video creator to synthesize a virtual scene as a storytelling background.

Recent advances in 3D content generation, ranging from SDS-based methods distilling 2D priors into neural fields (Poole et al., 2022; Wang et al., 2023; 2022; Tang et al., 2024b), to multi-view diffusion for geometric consistency (Liu et al., 2023b; Shi et al., 2023; 2024; Liu et al., 2024), and emerging 3D-native models like DiTs (Peebles & Xie, 2023; Hong et al., 2024b; Lai et al., 2025; Zhang et al., 2024b), have significantly improved 3D object synthesis. However, these methods are predominantly limited to single-object generation, with limited progress toward generating entire 3D scenes.

While directly generating 3D scenes is an active research direction, current scene-generation models tend to be either (1) domain-specific, trained on narrow datasets like indoor rooms or driving scenes; or (2) produce image-based representations such as panoramic or spherical projections, which are not designed for full 360-degree spatial interaction or view synthesis. This limitation arises from limited large-scale, diverse, general-purpose datasets for 3D scenes comparable to those available in 2D vision or object-centric 3D generation. Scenes contain long-range dependencies, heterogeneous object types, and complex spatial arrangements that are difficult to annotate and curate. Therefore, a method that can leverage object-centric generation to generate scenes is highly desired.

Recently, SynCity (Engstler et al., 2025) demonstrates promising results in adapting object generators for city-scale scene generation. However, its dependence on 2D inpainting (Lugmayr et al., 2022) introduces a fundamental limitation. Errors in the image domain can propagate and destabilize the 3D reconstruction. This raises a natural question: can we instead generate scenes directly in 3D space, bypassing the need for 2D intermediate representations?

In this work, we introduce **TRELLISWorld**, a training-free approach to text-driven 3D scene generation by leveraging general-purpose text-to-3D object diffusion models for scene composition. Our key insight is to reformulate global 3D scene synthesis as a multi-tile denoising problem, wherein the scene is partitioned into spatially overlapping regions that are independently denoised and later blended using a weighted averaging scheme in one diffusion step. This formulation offers a practical and scalable alternative to end-to-end scene-level training, enabling high-quality scene generation at significantly reduced cost. We will release the full implementation upon publication. Our method offers several advantages:

- **Training-free and editable**: Taking advantage of 3D environments' multi-scale signal structure, our approach requires no scene-level dataset or fine-tuning. It inherits editability and generalization capabilities from the underlying object-level generator, e.g., TRELLIS (Xiang et al., 2025).

- **Simple and general**: Our method requires minimal task-specific heuristics, making it broadly applicable across diverse scene types.

- **Scalable and smooth**: Compared to prior methods, our tile-wise approach is computationally efficient, blends overlapping regions smoothly, and enables the generation of significantly larger and more coherent scenes.

## 2 RELATED WORK

### 2.1 FOUNDATION OF RECONSTRUCTION AND OBJECT GENERATION

Benefiting from recent advances in 3D representations for reconstruction, such as NeRF (Mildenhall et al., 2020; Yu et al., 2021; Müller et al., 2022; Shue et al., 2022; Chen et al., 2022; Kerbl et al., 2023; Zhang et al., 2020; Barron et al., 2021; 2022; 2023), Score Distillation Sampling (SDS)-based methods (Poole et al., 2022; Wang et al., 2023; 2022; Tang et al., 2024b) distill the knowledge of 2D diffusion models into the creation of 3D objects using differentiable rendering. Specifically, NeRF++ (Zhang et al., 2020), combined with ProlificDreamer (Wang et al., 2023), demonstrates the first possibility of generating 3D scenes by distilling 2D diffusion with an added density prior. Later, multi-view diffusion methods (Liu et al., 2023b; Shi et al., 2023; 2024; Liu et al., 2024) were proposed to address the Janus (multi-face)

problem. However, as SDS depends on test-time optimization, object generation can take up to 40 minutes. To overcome this, methods like LRM have been proposed to generate 3D representations directly from images (Liu et al., 2023a; Hong et al., 2024b; Gao et al., 2024; Lai et al., 2025) and text (Jun & Nichol, 2023; Hong et al., 2024a; Zhang et al., 2024b). Although none of these works aim to generate large-scale scenes, their technologies lay the foundation for 3D scene generation.

## 2.2 Scene Generation Based on 2D Generation

While SDS-based methods above often focus on novel view synthesis or scene reconstruction, another line of work leverages the capabilities of 2D diffusion models without relying on test-time optimization. Early works extend a single image autoregressively into a video sequence, guided by camera trajectories (Liu et al., 2021; Li et al., 2022; Chai et al., 2023; Cai et al., 2023). These methods often project depth-estimated results into point clouds and fill in the missing regions under different camera extrinsics. While initial work focused mainly on natural scenes, the introduction of generic 2D diffusion models expanded the domain (Fridman et al., 2023; Shriram et al., 2025; Chung et al., 2023), and the integration of large language models (LLMs) enabled more diverse scene generation (Yu et al., 2024; 2025; Team et al., 2025). However, these works are limited to generating panoramic images or incomplete 3D representations that can only be viewed from a restricted set of camera extrinsics. A generic 3D generator capable of producing complete meshes remains necessary.

## 2.3 Scene Generation Based on 3D Scene-Native Generation

3D-native generation methods are promising alternatives to produce complete 3D representations, but they often rely on domain-specific training and are unable to generate generic scenes using natural language prompts (Lee et al., 2024; Tang et al., 2024a; Meng et al., 2025; Lee et al., 2025). For instance, works like InfiniCity (Lin et al., 2023; Xie et al., 2025b;a) depend heavily on RGB-D, semantic, or normal maps derived from satellite imagery. CityDreamer4D (Xie et al., 2025a) decomposes the generation task into multiple sub-tasks: layout, background, buildings, vehicles, and roads, each handled by a separate neural network. BlockFusion (Wu et al., 2024) uses autoregressive inference, while MIDI (Huang et al., 2025) employs multi-instance attention to scale object-level generation to scenes, both trained on dedicated 3D datasets (Fu et al., 2021; 2020). In contrast to publicly accessible 3D object datasets like Objaverse (Deitke et al., 2022; 2023), which contain over 10 million internet-sourced objects, the largest 3D scene dataset, FurniScene, contains only 100k rooms with 89 object classes (Zhang et al., 2024a). To the best of our knowledge, no openly available generic 3D scene dataset currently exists. Thus, a method that does not rely on curated 3D scene datasets but can still generate 3D representations across domains is highly desirable.

## 2.4 Scene Generation Based on 3D Object Generation

There are two categories of works that utilize 3D object generators to build scenes. The first category focuses on generating individual 3D objects and uses LLMs, visual-language models (VLMs), or image-based techniques to infer plausible object positions and orientations (Feng et al., 2023; Wu et al., 2025; Li et al., 2025). GALA3D (Zhou et al., 2024) employs SDS and additional physical losses to refine LLM-generated scene layouts from textual descriptions. CAST (Yao et al., 2025) constructs a relational graph, a constraint graph, and multiple masks from a single image to generate both object instances without occlusion and their spatial arrangement, using physical losses for consistency. However, these methods either depend on image inputs, limiting composition outside the view frustum, or rely on LLM reasoning, which often fails to produce accurate coordinates and complex inter-object relationships.

The second category relies on the inpainting capability of 3D diffusion models to compose and blend scenes from object-level generations. Most relevant to our work, SynCity (Engstler et al., 2025) generates object chunks autoregressively through a loop of 2D inpainting, 3D generation, and rendering. After generation, it fixes seams between chunks using 3D inpainting. To ensure consistency between 3D chunks and to avoid occlusions during 2D inpainting, multiple heuristics, such as cutting off parts of generated meshes for occlusion-free

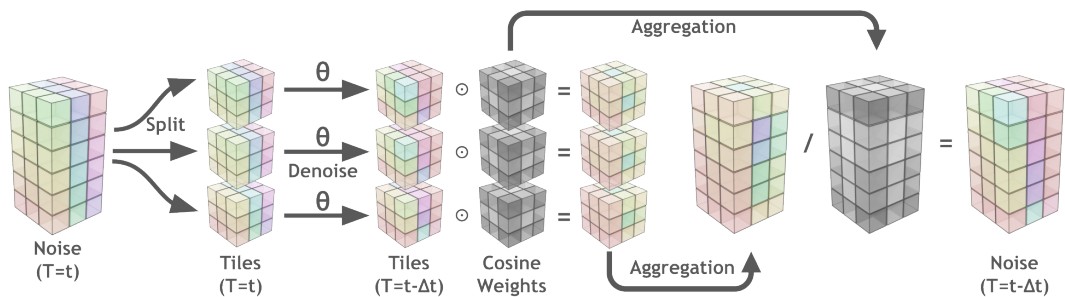

Figure 2: Illustration of our tiled diffusion process. We first split the scene noise into multiple tiles to denoise each tile in parallel. Then we take the weighted average described in Equation 1 for each tile and aggregate the result to obtain the scene noise for previous timesteps. This process is detailed in Equation 2.

renderings, are applied. However, these heuristics reduce generalizability beyond urban scenes and increase failure cases.

## 3 METHOD

The core of TRELLISWorld is a tiled diffusion with cosine blending. We first formulate the problem in subsection 3.1, then give a general method in subsection 3.2. We then describe how we implement this method using TRELLIS in subsection 3.3.

### 3.1 PROBLEM FORMULATION

Given a text-conditioned 3D generative diffusion model $\theta$ (a velocity field (Lipman et al., 2023)) that is capable of generating a 3D structure of size $S^3$ from a text prompt $p$, our goal is to generate a large-scale 3D world of arbitrary size $(X \times Y \times Z) \gg S^3$ that is consistent with the prompt. To simplify our explanation, without loss of generality, we assume $\theta$ is a pixel-diffusion model, meaning that the forward and reverse diffusion processes operate on the actual values instead of on compressed latents produced by autoencoders. Therefore, each object sample can be represented with a tensor in $\mathbb{R}^{S^3}$. We further assume that $\theta$ is trained on a general object distribution. To generate worlds using an object-level 3D generator, we leverage the local generation capability of $\theta$ while ensuring global coherence through careful conditioning and blending techniques.

### 3.2 TILED DIFFUSION

In this section, we demonstrate a simple method to convert a general 3D object generator into a general 3D scene generator.

We first initialize the entire world $W$ of size $(X, Y, Z)$ with Gaussian noise $W \sim \mathcal{N}(\mathbf{0}, \mathbf{I})$. We then divide the world into overlapping cubic tiles $\{w_i\}$ of size $(S, S, S)$ with a stride of $(s, s, s)$, where $s < S$ to ensure overlap between adjacent tiles. For a fixed scene, different settings of $s$ give different numbers of tiles. Thus, we use the word "chunk" to denote the absolute size of the scene. For each diffusion step, we process each tile $\{w_i\}$ in parallel and then aggregate the weighted average result to update the world $W$. The weight for each tile is defined by a 3D cosine mask that emphasizes the center of the tile and tapers off towards the edges, ensuring smooth transitions between adjacent tiles. See Figure 2 for visualization.

Formally, let $v^{(x,y,z)}$ be a voxel that has global position $(x, y, z)$ and let $f_{w_i} : \mathbb{N}^3 \to \mathbb{Z}^3$ be a function mapping global positions into local positions relative to tile $w_i$. Intuitively, if the resulting position $f_{w_i}(\cdot)$ lies in the set $\{0, ..., S-1\}^3$, then the tile $w_i$ covers the voxel at this input global position.

We define the weighting on local position $(x', y', z')$ to be:

$$\beta(x', y', z') = \begin{cases} \prod_{d \in \{x', y', z'\}} \cos\left(\pi\left(\frac{d+1}{S+1} - \frac{1}{2}\right)\right) & \text{if } (x', y', z') \in \{0, \ldots, S-1\}^3 \\ 0 & \text{otherwise} \end{cases} \quad (1)$$

Then, the update rule for each voxel $v^{(x,y,z)}$ in the world $W$ at diffusion step $t$ is given by the formula:

$$v_{t-\Delta t}^{(x,y,z)} = \frac{\sum_{\{w_i^{(t)}\}} \beta(f_{w_i^{(t)}}(x,y,z)) \cdot \left[w_i^{(t)} - \Delta t \cdot \theta(w_i^{(t)}, t) + \mathcal{O}(\Delta t^2)\right]_{f_{w_i^{(t)}}(x,y,z)}}{\sum_{\{w_i^{(t)}\}} \beta(f_{w_i^{(t)}}(x,y,z))} \quad (2)$$

where $\mathcal{O}(\Delta t^2)$ denotes the Big-O error resulting from Euler discretization, and the subscript $[\cdot]_{f_{w_i^{(t)}}(x,y,z)}$ denotes that $\theta$ operates in the local tile space.

### 3.3 Implementation

We build our method based on TRELLIS-text (Xiang et al., 2025), which is a text-conditioned diffusion transformer for 3D object generation. To perform object-level inference with TRELLIS, the process begins by denoising a $16^3$ noised latent using the *TRELLIS structure diffusion transformer* $\theta_1$. The resulting latent is decoded by *TRELLIS sparse structure decoder* to a $64^3$ occupancy grid (SS). This dense $64^3$ tensor is converted into a noisy sparse tensor by retaining only regions where the occupancy value exceeds zero. The noise $\sim \mathcal{N}(\mathbf{0}, \mathbf{I})$ is applied onto those regions. The sparse tensor is subsequently denoised using the *TRELLIS structure latent diffusion transformer* $\theta_2$, resulting in a structured latent representation (SLAT). Finally, the SLAT is decoded using the *TRELLIS structure latent Gaussian decoder* to produce a Gaussian Splatting representation (Kerbl et al., 2023).

**Tiled Diffusion** As described above, TRELLIS is a multi-stage latent diffusion model where $\theta_1$ operates on dense tensors and $\theta_2$ operates on sparse tensors. We perform tiled diffusion with minimal modification on both stages: the input tiles to $\theta_1, \theta_2$ are in encoded latent space and the corresponding masks are down-sampled by $4\times$. The blending and aggregation steps described in subsection 3.2 are performed in the latent space.

**Tiled Decoder** The resulting latent is decoded back to the voxel space and/or Gaussian Splatting space after the diffusion process is complete. Importantly, such decoding should also be done in a tiled manner. Since *TRELLIS structure latent Gaussian decoder* is not a probabilistic model, we set the stride $s = S$ to disable blending.

## 4 Experiments

We ablate our method in subsection 4.1 and compare blend quality, perceptual alignments, and computational cost with SynCity in subsection 4.2 and subsection 4.3. Unless otherwise noted, all experiments are conducted using classification-free guidance (Ho & Salimans, 2022) $cfg = 7.5$, stride $s = \frac{S}{2}$ and diffusion step size 25 on the Euler sampler.

### 4.1 Ablation

**Tiled Diffusion** A naive approach is to generate the world by stitching together multiple object generation results autoregressively, ensuring context alignment of neighboring chunks using inpainting. However, this leads to less coherent generation at the chunk edges, as shown in Figure 3.

**Tiled Decoder** We compare decoding using our tiled decoder with decoding the entire world at once in Figure 4. Removing the tiled decoder shows artifacts in the generated Gaussian Splatting.

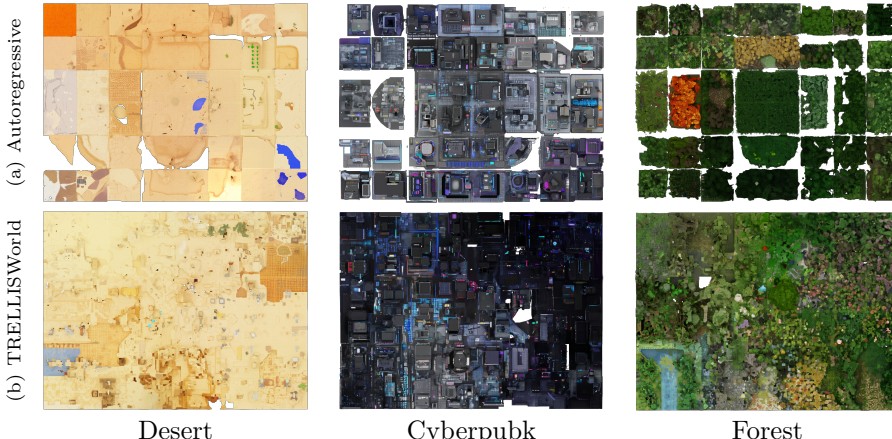

Figure 3: Top-down views of a generated 4x3x1 scene (not cherry-picked) using (a) an autoregressive method based on inpainting and (b) our method. Our method consistently shows better blending between tiles across different themes.

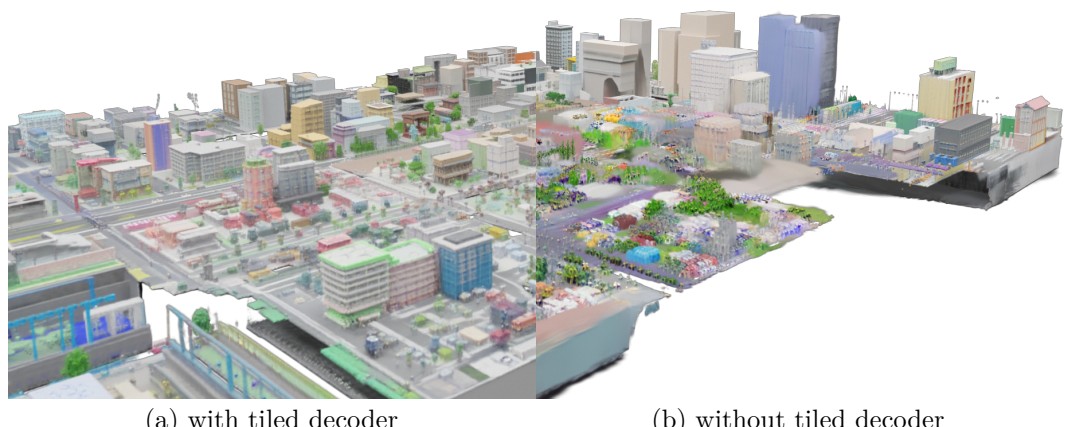

(a) with tiled decoder          (b) without tiled decoder

Figure 4: Comparison example with 3x2x1 city chunks for (a) decoding using our tiled decoding method and (b) decoding the entire generation at once. We observe severe artifacts when decoding without our tiled decoder.

**Blending**   We replace our blending method with a simple averaging aggregation. This results in visible seams across tile borders, as shown in Figure 5.

### 4.2 QUALITATIVE COMPARISON

We compare our method with SynCity (Engstler et al., 2025). Figure 6 showcases generation results across multiple prompts using the same 4x3x1 chunk layout. Our method tends to generate larger scenes and provides more natural blending across tiles compared to SynCity. For additional generation results of TRELLISWorld, see Figure 18 and Figure 19 in Appendix A.

Furthermore, our method is more robust compared to autoregressive methods based on image inpainting. For example, SynCity relies on heuristics such as cropping the 3D generation to avoid occlusion before applying 2D inpainting. Such heuristics are prone to failure when the 2D diffusion model mimics the cropped content, producing 3D chunks with artifacts, as shown in Figure 7.

### 4.3 QUANTITATIVE COMPARISON

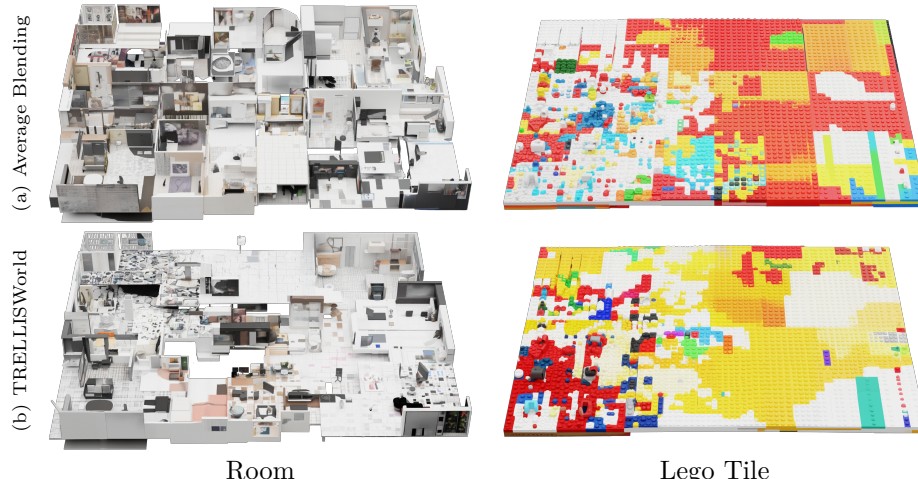

Room                                                    Lego Tile

Figure 5: Comparison (not cherry-picked) showing the effectiveness of blending. (a) With average blending, the "room" example tends to generate walls around tile borders, and the "lego tile" example produces a colored edge along tile borders, which is undesirable. (b) Tile borders become less noticeable with blending.

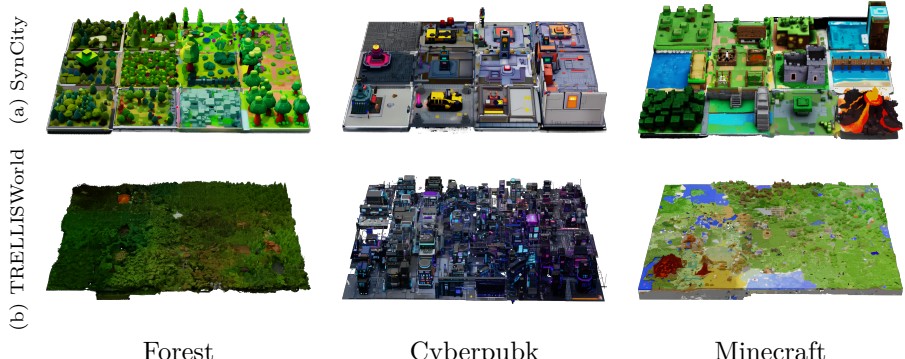

Forest                      Cyberpubk                      Minecraft

Figure 6: Qualitative comparison (not cherry-picked) between (a) SynCity (Engstler et al., 2025) and (b) TRELLISWorld (our method). All generations use a 4x3x1 chunk layout under the same Gaussian Splatting resolution. Our method demonstrates seamless blending between tiles, whereas chunk boundaries in SynCity are easily noticeable.

**Perceptual Alignments**  To compare perceptual alignments with SynCity using the CLIP score based on *clip-vit-base-patch32* model (Radford et al., 2021), we uniformly rendered 18 views at close distance from 15 generation results, each of size 4x3x1 across diverse prompts, visualized in Figure 14. We adopt the same set of prompts (e.g., "city", "medieval", "desert", ...) for both methods following the procedure detailed in subsection A.1. The result in Table 1 only shows marginal improvements, as the CLIP score does not directly measure blending quality across tiles.

**Seam Visibility**  The main contribution of our method lies in reducing seams between incoherent object generations. To measure the visibility of the seams, we first rendered the scene using orthographic projection and then calculated a binary map using the Canny edge detection operator. We use the average intensity of this binary map to define the seam visibility metric CannyAvg. You can find a detailed explanation of CannyAvg in subsubsection A.3.2.

**Computational Cost**  Without any optimization, TRELLISWorld takes 77.96 seconds on average to generate a chunk, which is a 5.80× speedup over SynCity (Engstler et al., 2025)

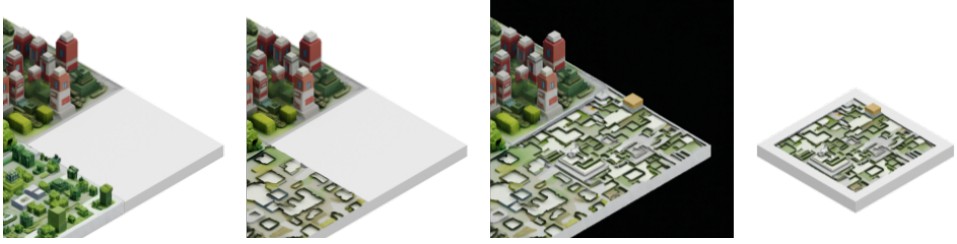

(a) previous tiles     (b) apply cropping     (b) inpainted image     (b) generated next tile

Figure 7: Limitations of world generation using autoregressive image inpainting methods described by SynCity (Engstler et al., 2025). In SynCity, to prevent tall buildings from previous chunks from occluding the next chunk, the previously generated result in (a) is cropped to (b), producing an inpainted image with artifacts in (c). The object generator is then conditioned on image (c) to generate Gaussian Splatting (d), which contains inherited artifacts.

Table 1: Perceptual scores and CannyAvg error with standard deviation and confidence intervals across different methods. Our proposed method achieves the highest CLIP (Radford et al., 2021) score and lowest CannyAvg error, indicating better alignment with users' prompts as well as minimal seams between tiles.

| Method | CLIP Mean ↑ | CLIP STD | CLIP 95% |
|---|---|---|---|
| SynCity | 0.26020 | 0.02322 | [0.25726, 0.26314] |
| inpaint baseline | 0.26419 | 0.02534 | [0.26099, 0.26740] |
| avg. blending | 0.26203 | 0.02704 | [0.25861, 0.26545] |
| TRELLISWorld | 0.26520 | 0.02496 | [0.26204, 0.26836] |
| | CannyAvg Mean ↑ | CannyAvg STD | CannyAvg 95% |
| SynCity | 7.81725 | 2.31850 | [6.98759, 8.64691] |
| inpaint baseline | 7.71797 | 5.63712 | [5.70075, 9.73519] |
| avg. blending | 6.55861 | 6.32146 | [4.29650, 8.82072] |
| TRELLISWorld | 5.61331 | 5.49558 | [3.64674, 7.57987] |

(452.04 seconds). TRELLISWorld also significantly reduces memory cost. During comparison to SynCity, we found that SynCity is unable to run on a single NVIDIA GeForce RTX 4080 (16GB). Therefore, we resort all SynCity experiments to a single NVIDIA GeForce RTX 4090 (48GB), a better GPU on all metrics, giving SynCity an unfair advantage. Figure 9 and Figure 10 show a linear increase in runtime as the number of chunks or tiles increases. Moreover, because our method does not rely on autoregressive inference, larger scenes can potentially be parallelized across multiple GPUs for further speedup.

## 5 Applications

**Editing or Expanding Existing Worlds** Our method can expand already-generated scenes by initializing the noise with parts of the ground truth, as shown in Figure 11 and detailed in subsection A.2.

**Area-Specific Prompting** Our method allows users to specify different prompts at each location. We demonstrate this capability in Figure 12. See subsection A.1 for details on creating area-specific prompts.

**Three-Dimensional Tiling** While most macro-structures on Earth are constrained to two-dimensional surfaces, our method naturally generalizes to the generation of three-dimensional macro-structures, such as a group of fish, or 3D blending using the area-specific prompting

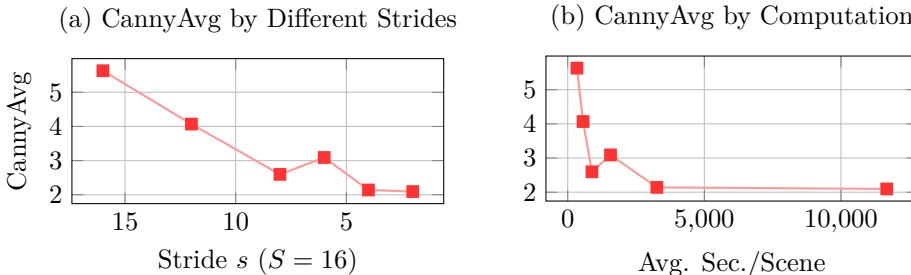

Figure 8: CannyAvg measuring seams in a scene using variable stride $s$. All experiments are conducted on a 4x3x1 chunk layout with a fixed prompt. Decreasing stride increases scene quality, with $s = 8$ being an optimal trade-off between computational time and scene quality.

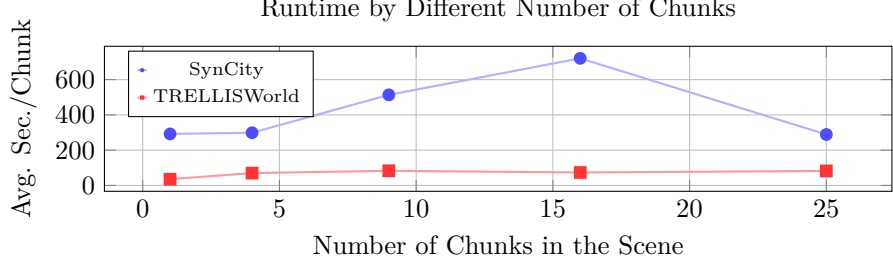

Figure 9: Computational cost required to generate a scene of variable scene sizes. All experiments are conducted on square chunk layouts from 1x1x1 to 5x5x1 with a fixed prompt. TRELLISWorld takes less time to generate a scene for all scene sizes.

technique described above, as shown in Figure 13. To our knowledge, no existing method offers this level of flexibility.

## 6 LIMITATIONS

While our method successfully generates coherent scenes without training, it presents several limitations for future investigations:

**Dependence on Base Models**  As a training-free approach, our method is inherently constrained by the capabilities of the underlying base models. In particular, the performance of TRELLIS directly limits both the visual fidelity and the efficiency of our scene generation pipeline.

**Object-Level Separation**  To ensure global scene coherence, our method performs generation in a single batch. As a result, it lacks the ability to disentangle individual objects post-generation.

## 7 CONCLUSION

We presented TRELLISWorld, a training-free framework for text-driven 3D scene generation that composes large-scale environments by repurposing object-level diffusion models through a tiled denoising formulation. By leveraging spatial overlap and cosine-weighted blending, our method enables semantically coherent, scalable, and editable 3D world synthesis without retraining. Experiments demonstrate that TRELLISWorld outperforms existing autoregressive approaches in both visual coherence and computational efficiency, while supporting flexible applications such as localized prompting. Our results establish a simple yet extensible

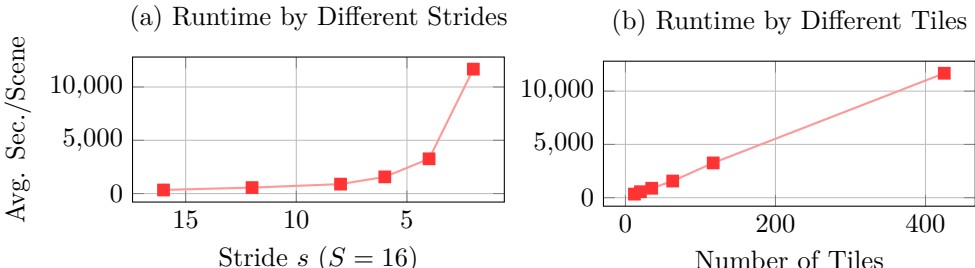

Figure 10: Computational cost required to generate a scene for variable stride $s$. All experiments are conducted on the 4x3x1 chunk layouts with a fixed prompt. Decreasing $s$ reciprocally increases the number of tiles, but the time to generate a tile remains constant (27.21 seconds per tile on average)

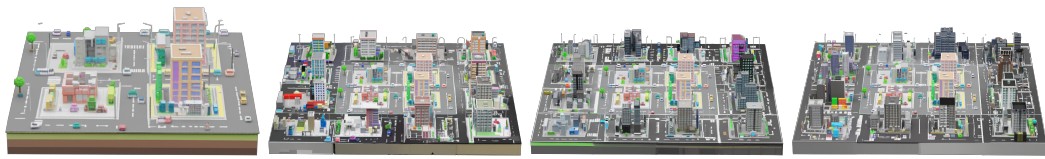

Figure 11: City scene expansion results (not cherry-picked) using TRELLISWorld. Given the leftmost 1x1x1 chunk as input, the model generates a 3x3x1 extended scene. Three diverse outputs are shown to the right, demonstrating variations.

foundation for general-purpose language-guided 3D scene construction, bridging the gap between object-level priors and world-scale generation.

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

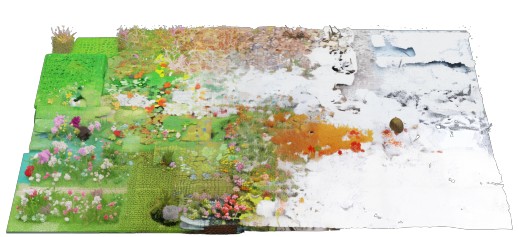

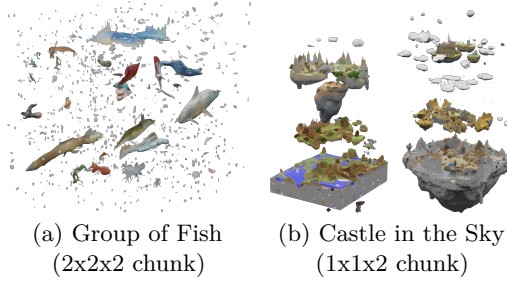

(a) Group of Fish (2x2x2 chunk)  (b) Castle in the Sky (1x1x2 chunk)

Figure 12: Generation result (not cherry-picked) showing a smooth and natural transition from "Spring forest tile... blooming flowers..." (bottom-left) to "Winter ice lake... skating marks..." (top-right).

Figure 13: Examples of three-dimensional tiling. The left scene is created using a uniform prompt: "A group of fish swimming in the air". The right scenes are two variations created using area-specific prompting.

Shengqu Cai, Eric Ryan Chan, Songyou Peng, Mohamad Shahbazi, Anton Obukhov, Luc Van Gool, and Gordon Wetzstein. Diffdreamer: Towards consistent unsupervised single-view scene extrapolation with conditional diffusion models, 2023. URL https://arxiv.org/abs/2211.12131.

Lucy Chai, Richard Tucker, Zhengqi Li, Phillip Isola, and Noah Snavely. Persistent nature: A generative model of unbounded 3d worlds, 2023. URL https://arxiv.org/abs/2303.13515.

Anpei Chen, Zexiang Xu, Andreas Geiger, Jingyi Yu, and Hao Su. Tensorf: Tensorial radiance fields, 2022. URL https://arxiv.org/abs/2203.09517.

Jaeyoung Chung, Suyoung Lee, Hyeongjin Nam, Jaerin Lee, and Kyoung Mu Lee. Luciddreamer: Domain-free generation of 3d gaussian splatting scenes, 2023. URL https://arxiv.org/abs/2311.13384.

Matt Deitke, Dustin Schwenk, Jordi Salvador, Luca Weihs, Oscar Michel, Eli VanderBilt, Ludwig Schmidt, Kiana Ehsani, Aniruddha Kembhavi, and Ali Farhadi. Objaverse: A universe of annotated 3d objects, 2022. URL https://arxiv.org/abs/2212.08051.

Matt Deitke, Ruoshi Liu, Matthew Wallingford, Huong Ngo, Oscar Michel, Aditya Kusupati, Alan Fan, Christian Laforte, Vikram Voleti, Samir Yitzhak Gadre, Eli VanderBilt, Aniruddha Kembhavi, Carl Vondrick, Georgia Gkioxari, Kiana Ehsani, Ludwig Schmidt, and Ali Farhadi. Objaverse-xl: A universe of 10m+ 3d objects, 2023. URL https://arxiv.org/abs/2307.05663.

Paul Engstler, Aleksandar Shtedritski, Iro Laina, Christian Rupprecht, and Andrea Vedaldi. Syncity: Training-free generation of 3d worlds, 2025. URL https://arxiv.org/abs/2503.16420.

Weixi Feng, Wanrong Zhu, Tsu jui Fu, Varun Jampani, Arjun Akula, Xuehai He, Sugato Basu, Xin Eric Wang, and William Yang Wang. Layoutgpt: Compositional visual planning and generation with large language models, 2023. URL https://arxiv.org/abs/2305.15393.

Rafail Fridman, Amit Abecasis, Yoni Kasten, and Tali Dekel. Scenescape: Text-driven consistent scene generation, 2023. URL https://arxiv.org/abs/2302.01133.

Huan Fu, Rongfei Jia, Lin Gao, Mingming Gong, Binqiang Zhao, Steve Maybank, and Dacheng Tao. 3d-future: 3d furniture shape with texture, 2020. URL https://arxiv.org/abs/2009.09633.

Huan Fu, Bowen Cai, Lin Gao, Lingxiao Zhang, Jiaming Wang Cao Li, Zengqi Xun, Chengyue Sun, Rongfei Jia, Binqiang Zhao, and Hao Zhang. 3d-front: 3d furnished rooms with layouts and semantics, 2021. URL https://arxiv.org/abs/2011.09127.

Ruiqi Gao, Aleksander Holynski, Philipp Henzler, Arthur Brussee, Ricardo Martin-Brualla, Pratul Srinivasan, Jonathan T. Barron, and Ben Poole. Cat3d: Create anything in 3d with multi-view diffusion models, 2024. URL https://arxiv.org/abs/2405.10314.

Jonathan Ho and Tim Salimans. Classifier-free diffusion guidance, 2022. URL https://arxiv.org/abs/2207.12598.

Fangzhou Hong, Jiaxiang Tang, Ziang Cao, Min Shi, Tong Wu, Zhaoxi Chen, Shuai Yang, Tengfei Wang, Liang Pan, Dahua Lin, and Ziwei Liu. 3dtopia: Large text-to-3d generation model with hybrid diffusion priors, 2024a. URL https://arxiv.org/abs/2403.02234.

Yicong Hong, Kai Zhang, Jiuxiang Gu, Sai Bi, Yang Zhou, Difan Liu, Feng Liu, Kalyan Sunkavalli, Trung Bui, and Hao Tan. Lrm: Large reconstruction model for single image to 3d, 2024b. URL https://arxiv.org/abs/2311.04400.

Zehuan Huang, Yuan-Chen Guo, Xingqiao An, Yunhan Yang, Yangguang Li, Zi-Xin Zou, Ding Liang, Xihui Liu, Yan-Pei Cao, and Lu Sheng. Midi: Multi-instance diffusion for single image to 3d scene generation, 2025. URL https://arxiv.org/abs/2412.03558.

Heewoo Jun and Alex Nichol. Shap-e: Generating conditional 3d implicit functions, 2023. URL https://arxiv.org/abs/2305.02463.

Bernhard Kerbl, Georgios Kopanas, Thomas Leimkühler, and George Drettakis. 3d gaussian splatting for real-time radiance field rendering, 2023. URL https://arxiv.org/abs/2308.04079.

Zeqiang Lai, Yunfei Zhao, Haolin Liu, Zibo Zhao, Qingxiang Lin, Huiwen Shi, Xianghui Yang, Mingxin Yang, Shuhui Yang, Yifei Feng, Sheng Zhang, Xin Huang, Di Luo, Fan Yang, Fang Yang, Lifu Wang, Sicong Liu, Yixuan Tang, Yulin Cai, Zebin He, Tian Liu, Yuhong Liu, Jie Jiang, Linus, Jingwei Huang, and Chunchao Guo. Hunyuan3d 2.5: Towards high-fidelity 3d assets generation with ultimate details, 2025. URL https://arxiv.org/abs/2506.16504.

Han-Hung Lee, Qinghong Han, and Angel X. Chang. Nuiscene: Exploring efficient generation of unbounded outdoor scenes, 2025. URL https://arxiv.org/abs/2503.16375.

Jumin Lee, Sebin Lee, Changho Jo, Woobin Im, Juhyeong Seon, and Sung-Eui Yoon. Semcity: Semantic scene generation with triplane diffusion, 2024. URL https://arxiv.org/abs/2403.07773.

Qixuan Li, Chao Wang, Zongjin He, and Yan Peng. Phip-g: Physics-guided text-to-3d compositional scene generation, 2025. URL https://arxiv.org/abs/2502.00708.

Zhengqi Li, Qianqian Wang, Noah Snavely, and Angjoo Kanazawa. Infinitenature-zero: Learning perpetual view generation of natural scenes from single images, 2022. URL https://arxiv.org/abs/2207.11148.

Chieh Hubert Lin, Hsin-Ying Lee, Willi Menapace, Menglei Chai, Aliaksandr Siarohin, Ming-Hsuan Yang, and Sergey Tulyakov. Infinicity: Infinite-scale city synthesis, 2023. URL https://arxiv.org/abs/2301.09637.

Yaron Lipman, Ricky T. Q. Chen, Heli Ben-Hamu, Maximilian Nickel, and Matt Le. Flow matching for generative modeling, 2023. URL https://arxiv.org/abs/2210.02747.

Andrew Liu, Richard Tucker, Varun Jampani, Ameesh Makadia, Noah Snavely, and Angjoo Kanazawa. Infinite nature: Perpetual view generation of natural scenes from a single image, 2021. URL https://arxiv.org/abs/2012.09855.

Minghua Liu, Chao Xu, Haian Jin, Linghao Chen, Mukund Varma T, Zexiang Xu, and Hao Su. One-2-3-45: Any single image to 3d mesh in 45 seconds without per-shape optimization, 2023a. URL https://arxiv.org/abs/2306.16928.

Ruoshi Liu, Rundi Wu, Basile Van Hoorick, Pavel Tokmakov, Sergey Zakharov, and Carl Vondrick. Zero-1-to-3: Zero-shot one image to 3d object, 2023b. URL https://arxiv.org/abs/2303.11328.

Yuan Liu, Cheng Lin, Zijiao Zeng, Xiaoxiao Long, Lingjie Liu, Taku Komura, and Wenping Wang. Syncdreamer: Generating multiview-consistent images from a single-view image, 2024. URL https://arxiv.org/abs/2309.03453.

Andreas Lugmayr, Martin Danelljan, Andres Romero, Fisher Yu, Radu Timofte, and Luc Van Gool. Repaint: Inpainting using denoising diffusion probabilistic models, 2022. URL https://arxiv.org/abs/2201.09865.

Yanxu Meng, Haoning Wu, Ya Zhang, and Weidi Xie. Scenegen: Single-image 3d scene generation in one feedforward pass, 2025. URL https://arxiv.org/abs/2508.15769.

Ben Mildenhall, Pratul P. Srinivasan, Matthew Tancik, Jonathan T. Barron, Ravi Ramamoorthi, and Ren Ng. Nerf: Representing scenes as neural radiance fields for view synthesis, 2020. URL https://arxiv.org/abs/2003.08934.

Thomas Müller, Alex Evans, Christoph Schied, and Alexander Keller. Instant neural graphics primitives with a multiresolution hash encoding. *ACM Transactions on Graphics*, 41(4):1–15, July 2022. ISSN 1557-7368. doi: 10.1145/3528223.3530127. URL http://dx.doi.org/10.1145/3528223.3530127.

William Peebles and Saining Xie. Scalable diffusion models with transformers, 2023. URL https://arxiv.org/abs/2212.09748.

Ben Poole, Ajay Jain, Jonathan T. Barron, and Ben Mildenhall. Dreamfusion: Text-to-3d using 2d diffusion, 2022. URL https://arxiv.org/abs/2209.14988.

Alec Radford, Jong Wook Kim, Chris Hallacy, Aditya Ramesh, Gabriel Goh, Sandhini Agarwal, Girish Sastry, Amanda Askell, Pamela Mishkin, Jack Clark, Gretchen Krueger, and Ilya Sutskever. Learning transferable visual models from natural language supervision, 2021. URL https://arxiv.org/abs/2103.00020.

Ruoxi Shi, Hansheng Chen, Zhuoyang Zhang, Minghua Liu, Chao Xu, Xinyue Wei, Linghao Chen, Chong Zeng, and Hao Su. Zero123++: a single image to consistent multi-view diffusion base model, 2023. URL https://arxiv.org/abs/2310.15110.

Yichun Shi, Peng Wang, Jianglong Ye, Mai Long, Kejie Li, and Xiao Yang. Mvdream: Multi-view diffusion for 3d generation, 2024. URL https://arxiv.org/abs/2308.16512.

Jaidev Shriram, Alex Trevithick, Lingjie Liu, and Ravi Ramamoorthi. Realmdreamer: Text-driven 3d scene generation with inpainting and depth diffusion, 2025. URL https://arxiv.org/abs/2404.07199.

J. Ryan Shue, Eric Ryan Chan, Ryan Po, Zachary Ankner, Jiajun Wu, and Gordon Wetzstein. 3d neural field generation using triplane diffusion, 2022. URL https://arxiv.org/abs/2211.16677.

Jiapeng Tang, Yinyu Nie, Lev Markhasin, Angela Dai, Justus Thies, and Matthias Nießner. Diffuscene: Denoising diffusion models for generative indoor scene synthesis, 2024a. URL https://arxiv.org/abs/2303.14207.

Jiaxiang Tang, Jiawei Ren, Hang Zhou, Ziwei Liu, and Gang Zeng. Dreamgaussian: Generative gaussian splatting for efficient 3d content creation, 2024b. URL https://arxiv.org/abs/2309.16653.

HunyuanWorld Team, Zhenwei Wang, Yuhao Liu, Junta Wu, Zixiao Gu, Haoyuan Wang, Xuhui Zuo, Tianyu Huang, Wenhuan Li, Sheng Zhang, Yihang Lian, Yulin Tsai, Lifu Wang, Sicong Liu, Puhua Jiang, Xianghui Yang, Dongyuan Guo, Yixuan Tang, Xinyue Mao, Jiaao Yu, Junlin Yu, Jihong Zhang, Meng Chen, Liang Dong, Yiwen Jia, Chao Zhang, Yonghao Tan, Hao Zhang, Zheng Ye, Peng He, Runzhou Wu, Minghui Chen, Zhan Li, Wangchen Qin, Lei Wang, Yifu Sun, Lin Niu, Xiang Yuan, Xiaofeng Yang, Yingping He, Jie Xiao, Yangyu Tao, Jianchen Zhu, Jinbao Xue, Kai Liu, Chongqing Zhao, Xinming Wu, Tian Liu, Peng Chen, Di Wang, Yuhong Liu, Linus, Jie Jiang, Tengfei Wang, and Chunchao Guo. Hunyuanworld 1.0: Generating immersive, explorable, and interactive 3d worlds from words or pixels, 2025. URL https://arxiv.org/abs/2507.21809.

Haochen Wang, Xiaodan Du, Jiahao Li, Raymond A. Yeh, and Greg Shakhnarovich. Score jacobian chaining: Lifting pretrained 2d diffusion models for 3d generation, 2022. URL https://arxiv.org/abs/2212.00774.

Zhengyi Wang, Cheng Lu, Yikai Wang, Fan Bao, Chongxuan Li, Hang Su, and Jun Zhu. Prolificdreamer: High-fidelity and diverse text-to-3d generation with variational score distillation, 2023. URL https://arxiv.org/abs/2305.16213.

Qirui Wu, Denys Iliash, Daniel Ritchie, Manolis Savva, and Angel X. Chang. Diorama: Unleashing zero-shot single-view 3d indoor scene modeling, 2025. URL https://arxiv.org/abs/2411.19492.

Zhennan Wu, Yang Li, Han Yan, Taizhang Shang, Weixuan Sun, Senbo Wang, Ruikai Cui, Weizhe Liu, Hiroyuki Sato, Hongdong Li, and Pan Ji. Blockfusion: Expandable 3d scene generation using latent tri-plane extrapolation, 2024. URL https://arxiv.org/abs/2401.17053.

Jianfeng Xiang, Zelong Lv, Sicheng Xu, Yu Deng, Ruicheng Wang, Bowen Zhang, Dong Chen, Xin Tong, and Jiaolong Yang. Structured 3d latents for scalable and versatile 3d generation, 2025. URL `https://arxiv.org/abs/2412.01506`.

Haozhe Xie, Zhaoxi Chen, Fangzhou Hong, and Ziwei Liu. Compositional generative model of unbounded 4d cities. *IEEE Transactions on Pattern Analysis and Machine Intelligence*, pp. 1–17, 2025a. ISSN 1939-3539. doi: 10.1109/tpami.2025.3603078. URL `http://dx.doi.org/10.1109/TPAMI.2025.3603078`.

Haozhe Xie, Zhaoxi Chen, Fangzhou Hong, and Ziwei Liu. Generative gaussian splatting for unbounded 3d city generation, 2025b. URL `https://arxiv.org/abs/2406.06526`.

Kaixin Yao, Longwen Zhang, Xinhao Yan, Yan Zeng, Qixuan Zhang, Wei Yang, Lan Xu, Jiayuan Gu, and Jingyi Yu. Cast: Component-aligned 3d scene reconstruction from an rgb image, 2025. URL `https://arxiv.org/abs/2502.12894`.

Alex Yu, Sara Fridovich-Keil, Matthew Tancik, Qinhong Chen, Benjamin Recht, and Angjoo Kanazawa. Plenoxels: Radiance fields without neural networks, 2021. URL `https://arxiv.org/abs/2112.05131`.

Hong-Xing Yu, Haoyi Duan, Junhwa Hur, Kyle Sargent, Michael Rubinstein, William T. Freeman, Forrester Cole, Deqing Sun, Noah Snavely, Jiajun Wu, and Charles Herrmann. Wonderjourney: Going from anywhere to everywhere, 2024. URL `https://arxiv.org/abs/2312.03884`.

Hong-Xing Yu, Haoyi Duan, Charles Herrmann, William T. Freeman, and Jiajun Wu. Wonderworld: Interactive 3d scene generation from a single image, 2025. URL `https://arxiv.org/abs/2406.09394`.

Genghao Zhang, Yuxi Wang, Chuanchen Luo, Shibiao Xu, Zhaoxiang Zhang, Man Zhang, and Junran Peng. Furniscene: A large-scale 3d room dataset with intricate furnishing scenes, 2024a. URL `https://arxiv.org/abs/2401.03470`.

Kai Zhang, Gernot Riegler, Noah Snavely, and Vladlen Koltun. Nerf++: Analyzing and improving neural radiance fields, 2020. URL `https://arxiv.org/abs/2010.07492`.

Longwen Zhang, Ziyu Wang, Qixuan Zhang, Qiwei Qiu, Anqi Pang, Haoran Jiang, Wei Yang, Lan Xu, and Jingyi Yu. Clay: A controllable large-scale generative model for creating high-quality 3d assets, 2024b. URL `https://arxiv.org/abs/2406.13897`.

Xiaoyu Zhou, Xingjian Ran, Yajiao Xiong, Jinlin He, Zhiwei Lin, Yongtao Wang, Deqing Sun, and Ming-Hsuan Yang. Gala3d: Towards text-to-3d complex scene generation via layout-guided generative gaussian splatting, 2024. URL `https://arxiv.org/abs/2402.07207`.

## A  APPENDIX

### A.1  TEXT PROMPT

Our method gives the user explicit control of the 3D prompts used in generation. Below is an example of a 3D prompt we use to generate our 4x3x1 city theme:

```
prompt = [[
    ["Dense low-rise residential block with small shops and narrow
        streets, square tile"],
    ["Cluster of mid-rise apartments with pastel facades and tree-
        lined sidewalks, square tile"],
    ["Modern commercial zone with glass offices, cafes, and public
        seating, square tile"],
], [
    ["Mixed-use area with offices, apartments, and green courtyards,
        square tile"],
    ["Urban block with mid-rise towers, parking lots, and small plazas
        , square tile"],
```

```
    ["Dense retail and commercial buildings near busy intersection,
        square tile"],
], [
    ["Residential zone with consistent low-rise buildings and local
        shops, square tile"],
    ["Compact city block with modern mid-rises and organized street
        grid, square tile"],
    ["Edge of city with fewer high-rises and more greenery, square
        tile"],
], [
    ["Park extension with dense trees and a water feature, square tile
        "],
    ["Community recreation area with playgrounds and open lawns,
        square tile"],
    ["Park transition with scattered cultural buildings and trees,
        square tile"],
]]
```

For other themes ("city", "medieval", "desert", "cyberpunk", "ancient Rome", "minecraft", "forest", "ocean", "winter", "lego", "park", "amusement park", "airport", "college", "room"), we ask LLMs to generate similar prompts using in-context learning from the city prompt above. For example, we generated our medieval prompt by asking "[IN_CONTEXT_PROMPT] Above is a prompt for creating a large city block with parks, please give a prompt for 'a medieval tile with farmland fields and cottages' following the same format."

As shown above, the input prompt is formatted as a 3D tensor. Therefore, prompts in spatial proximity will remain close in the generated scene. We do not force the user to provide a tensor that has the exact shape of our tiles, as this may limit flexibility and ease of use. Instead, we treat the input prompt tensor as a tensor spanning the entire scene. Specifically, given a tile location, we sample the nearest prompt in the prompt tensor. This way, users can provide a coarser prompt tensor.

### A.2 Inpaint

Our method can fill missing regions or extend a user-provided chunks. We use RePaint (Lugmayr et al., 2022) with a Gaussian-blurred mask to partially preserve the edges of the input chunks. This encourages smoother transitions between the generated and existing contents.

### A.3 Metrics

#### A.3.1 CLIP

We measured CLIP similarity using 18 views per generated scene. Figure 14 shows 6 different views from SynCity and TRELLISWorld respectively. We used the mega-prompt (e.g., 'a medieval tile with farmland fields and cottages') instead of the object-level prompt for CLIP text input, appending ", brid eye view" and removing all auxiliary modifications such as ", on top of a base" and ", square tile" used by SynCity and TRELLISWorld to better reflect the actual content in the images.

#### A.3.2 CannyAvg

We used CannyAvg to evaluate the amount of seams in our generated results. The CannyAvg value for each scene is calculated as the average pixel intensity of an orthographic top-down rendering after Canny edge detection with a $[200, 400]$ threshold. For each 4x3x1 generated result, we rendered an image at 1536x2048 resolution. To show the effectiveness of CannyAvg, we have included Canny edge detection results in Figure 15, along with other possible edge detection methods that are less effective at highlighting seams.

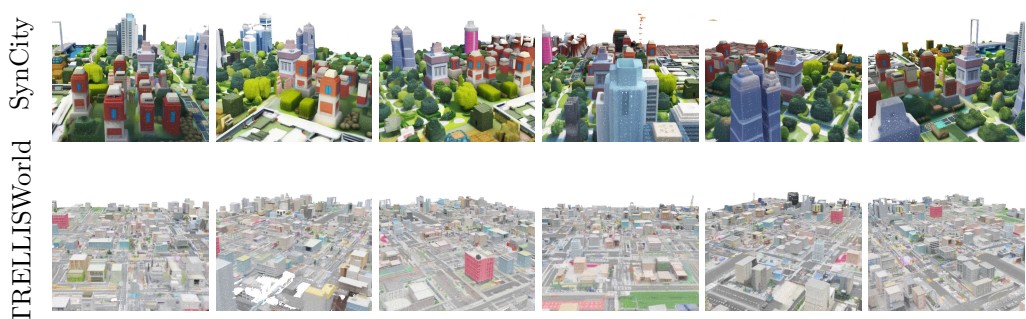

Figure 14: Sampled views for CLIP similarity calculation.

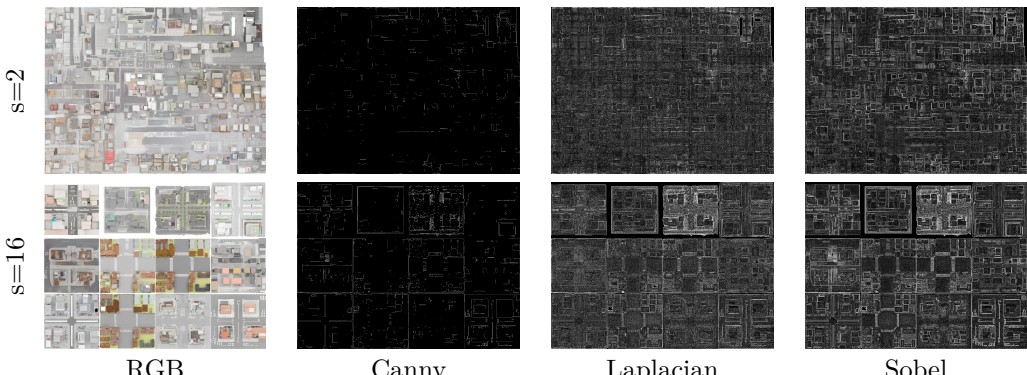

Figure 15: Visualization of different edge detection methods. Canny best aligns with human perception of seams, while other methods highlight internal content, making them unsuitable for quantitative seam measurement.

### A.4 EXTENSION TO OTHER BASE GENERATOR

To demonstrate the extensiveness of our method, we have tested our method using two other base generators. The first is an unconditional diffusion transformer trained on Minecraft terrains and the second is an image-conditioned triplane diffusion transformer trained on crops of Minecraft structures. While both architectures only support generation of 16x16x16 blocks layout, we adapt them using our method to generate structures with sizes beyond what they were trained on. The qualitative results in Figure 16 shows that our method keeps the generation cohesive without any visible seams.

### A.5 EXTENSION TO IMAGE-CONDITIONED GENERATOR

We argue that our method is best applied to a text-conditioned object generator. Applying our method to image-conditioned object generators often result in unaligned floor level, which is an indicator of unsuccessful blending. We show generation results using the image-conditioned TRELLIS model in Figure 17.

We suspect that this discrepancy between text-conditioned and image-conditioned model lies in the difference in their underlying learned conditional distributions of the score models $\theta_{img}, \theta_{text}$. When the model $\theta_{img}, \theta_{text}$ is trained on the same set of 3D object but with different conditional label $c_{img}, c_{text}$. They have the same marginalized distribution $q_{\theta_{img}}(x_0) = \int q_{\theta_{img}}(x_0 \mid c_{img})p(c_{img})dc_{img} = \int q_{\theta_{text}}(x_0 \mid c_{text})p(c_{text})dc_{text} = q_{\theta_{text}}(x_0)$. However, their conditional distribution is different: $q_{\theta_{text}}(x_0 \mid c_{text} = \text{specific-text}) \neq q_{\theta_{img}}(x_0 \mid c_{img} = \text{specific-img})$ even when the given "specific-text" and "specific-img" are associated with the same 3D object. Specifically, $q_{\theta_{text}}(x_0 \mid c_{text})$ might be more diffuse

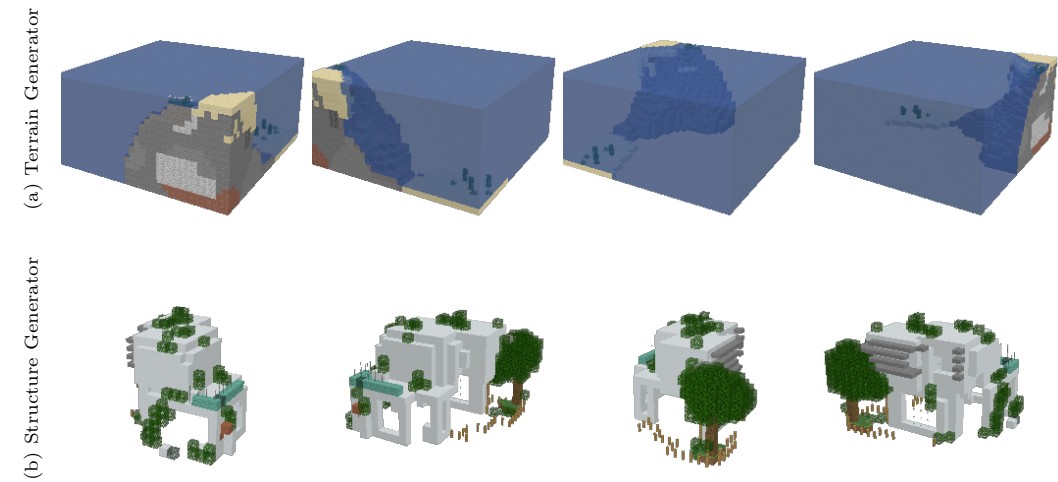

Figure 16: Generation results of tiled diffusion applied to two different Minecraft object generators. (a) The Minecraft terrain generator is trained on 16x16x16 blocks but sampled at 32x32x16 blocks. (b) The Minecraft structure generator is trained on 16x16x16 blocks but sampled at 22x21x14 blocks.

compared to $q_{\theta_{img}}(x_0 \mid c_{img})$, which makes the text-conditioned model more robust to out-of-data-distribution queries of the score function.

### A.6 ADDITIONAL QUALITATIVE RESULTS

Figure 18 and Figure 19 shows additional generation result of TRELLISWorld.

### A.7 DISCLOSURE

We made use of LLMs to polish writing. We made sure that our input text to LLMs will not be used for training purposes.

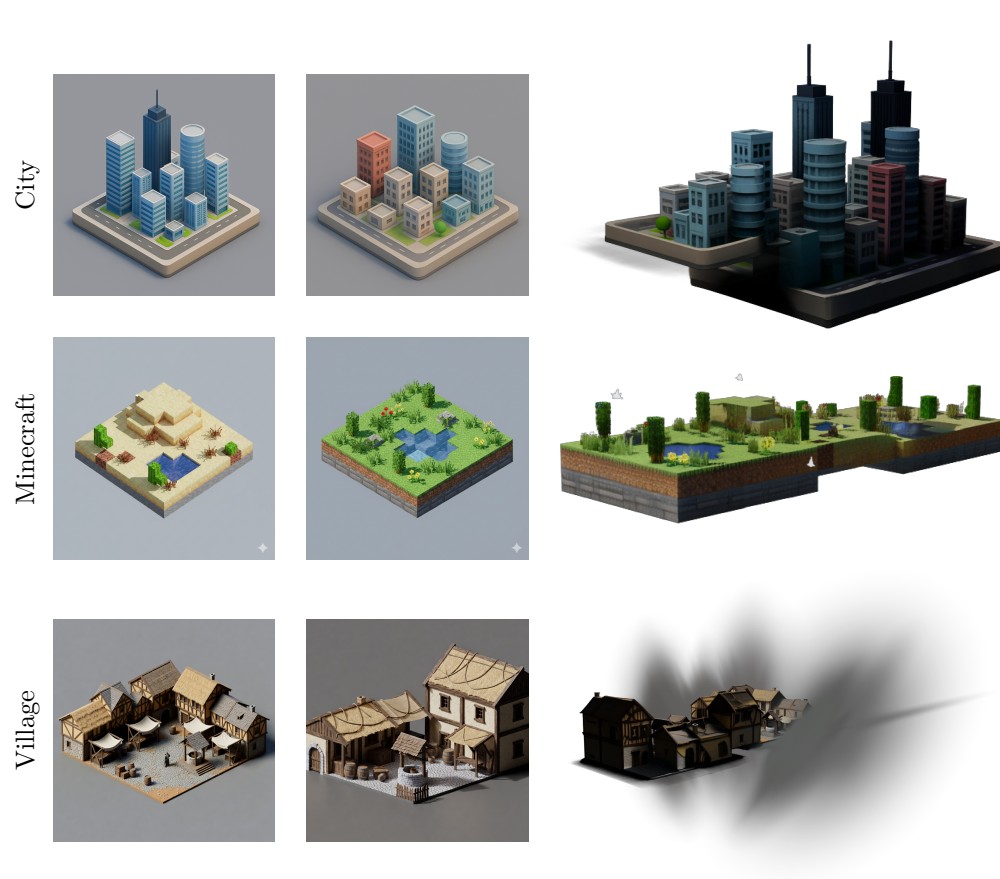

Image Left          Image Right          Generated Result

Figure 17: Visualization of applying TRELLISWorld to image-conditioned generator. We generated 2x1x1 chunk which was split to 3 tiles (left, right, and middle tile) from 2 conditional images. The middle image was chosen arbitrarily from left or right images. In all generations give inconsistent floor levels. The "Village" example shows significant artifact.

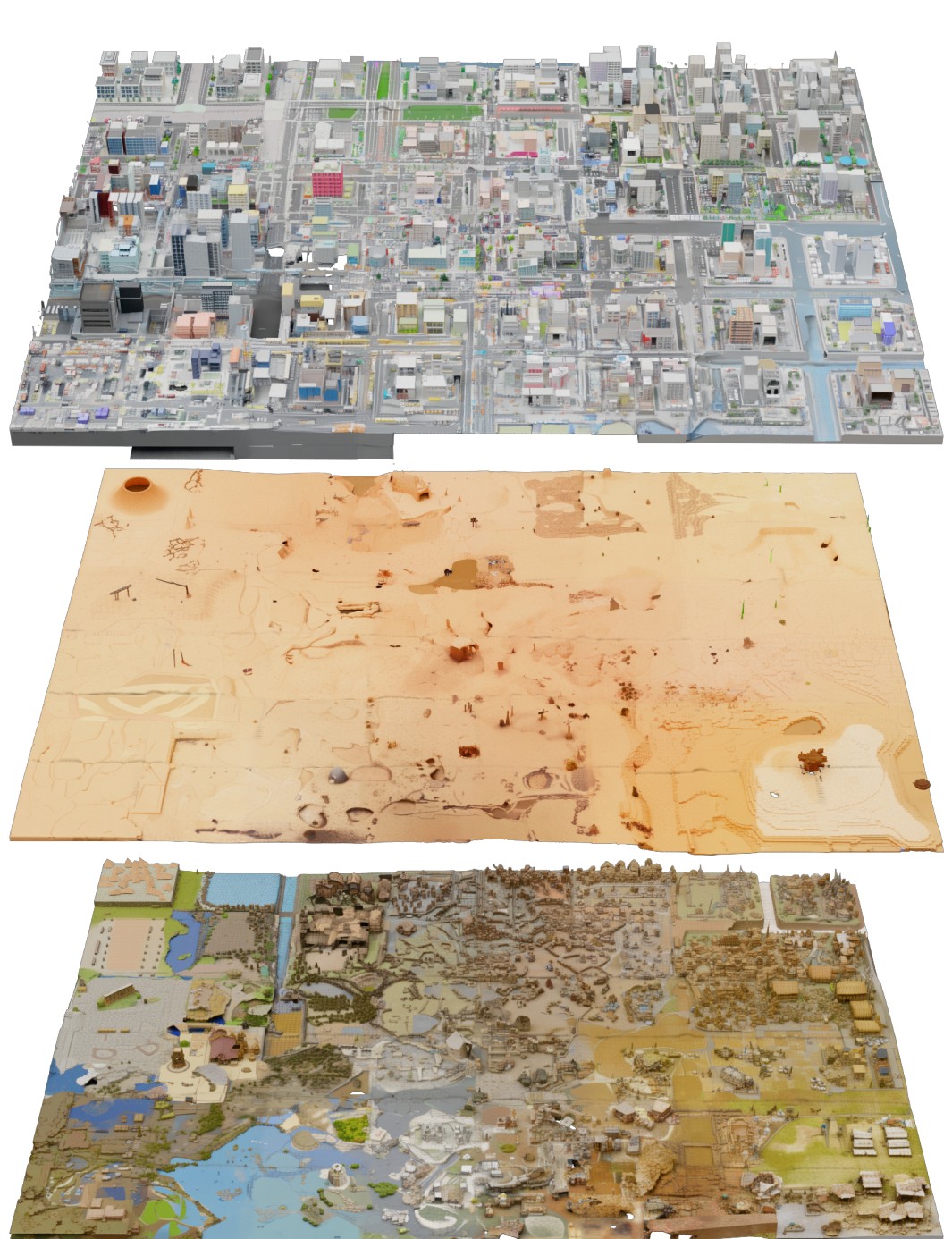

Figure 18: Additional (not cherry-picked) 4x3x1 chunk Tiled Gaussian Splatting Generation Results: city, desert, medieval.

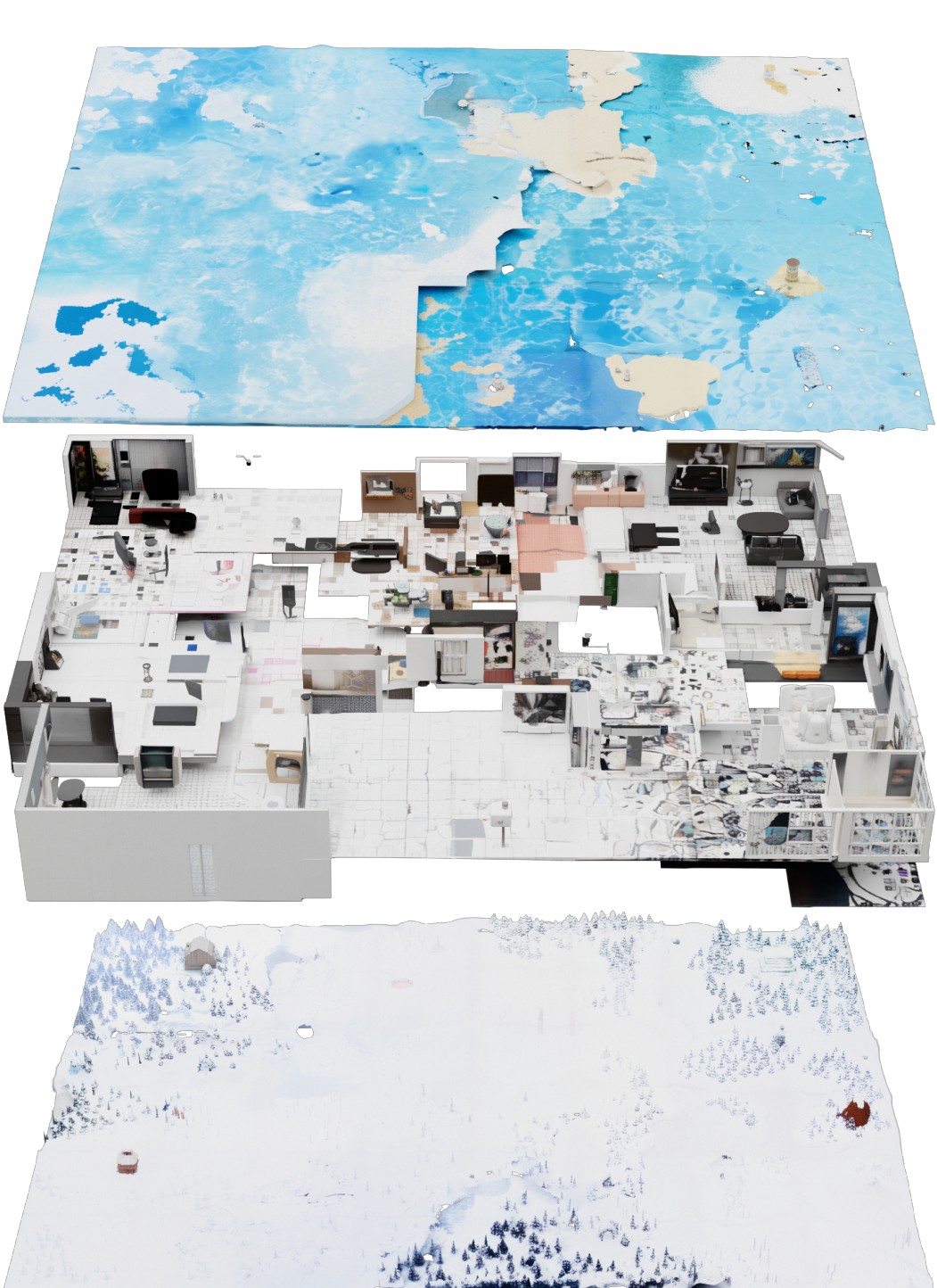

Figure 19: Additional (not cherry-picked) 4x3x1 chunk Tiled Gaussian Splatting Generation Results: ocean, room, winter.

