# OpenReview forum: "TRELLISWorld: Training-Free World Generation from Object Generators"
_ICLR.cc/2026/Conference — Submitted to ICLR 2026_

### Official Review · Reviewer_7XGV · 2025-10-29

**Soundness:** 3
**Presentation:** 3
**Contribution:** 2
**Rating:** 4
**Confidence:** 4

**Summary:**

This paper introduces TRELLISWorld, a training-free framework for large-scale 3D scene generation. The main contribution is that it enables scene-level generation without requiring scene-level training data, leveraging object-level pretrained 3D generative models (Trellis) instead.
The method divides a 3D scene into multiple overlapping patches. For patches with shared regions, a weighted averaging over the overlapping areas is applied during generation to ensure local consistency. Ablation studies show that using a 3D cosine mask for blending produces smoother transitions than direct averaging.
Compared with SynCity, TRELLISWorld achieves a slight improvement on CLIP Mean, but at the cost of significantly higher generation time and computational resources.

**Strengths:**

1. Training-free and data-efficient: The method does not rely on scene-level 3D data; it extends an object-level pretrained model to generate full 3D scenes.
2. Effective patch blending: The use of a 3D cosine mask during blending helps preserve the central region of each generated patch while enabling smooth transitions at the edges.

**Weaknesses:**

1. Evaluation inconsistency: For perceptual alignment comparison, the paper states that 18 close-distance views were uniformly sampled, yet the presented examples are mostly long-distance views. Showing close-up comparisons would provide a more convincing evaluation.
2. Limited controllability: The method only supports text-based control, which limits fine-grained scene manipulation. Designing prompts for each patch can be cumbersome, even though the authors mention using LLMs to generate multiple prompts. However, how these prompts are assigned to specific patches to ensure coherent spatial layout is unclear.
3. Low-resolution figures: The figures are generally low in resolution, making it difficult to assess the fine-grained 3D details of generated objects.

**Questions:**

1. The overall quality of the results is hard to assess since most examples only show low-resolution overviews of entire scenes. It is strongly recommended that the authors include close-up visualizations and higher-resolution renderings to better demonstrate scene quality.
2. As the method relies on Trellis, a text-to-3D model, each patch requires a corresponding text prompt. However, the paper lacks a detailed explanation of how prompts are managed or distributed across patches (e.g., after generating multiple city-related prompts via LLMs, how are they spatially allocated to form a coherent scene?).

Things to improve the paper that did not impact the score:
- Figure 2 should be moved to page 4, closer to where it is referenced.
- In Formula (2), please clarify the meaning of the $\mathcal{O}$ notation and $[\cdot]_f$.
- For a tile size of 4×3×1, what is S? In Figure 3(a), it seems the layout is divided into 8×6 tiles. So is S = 0.5?
- In Figure 4, “without blending” is misleading since simple averaging aggregation is also a form of blending. Consider using “average” and “weighted average” or others instead.
- The tiled decoder mentioned in the ablation study is difficult to locate in the Method section; please reorganize the text to highlight this component more clearly.

---

> ### Author Response · Authors · 2025-11-22
>
> Dear Reviewer 7XGV,
>
> Thank you for your thoughtful review and positive comments regarding our submission. As you mentioned, compared to SynCity, we indeed achieve a slightly better CLIP score. However, this is *without* compromising computational efficiency. Notably, our method is significantly more efficient, requiring only about 20\% of the computational resources compared to SynCity, while still achieving superior scene quality. We believe this efficiency gain is a crucial contribution of our work.
>
> Incorporating feedback from peers, we realized that our clear improvement in visual quality does not transfer to CLIP score visibly enough as CLIP does not decrease significantly even with visible seams in the scene. To this end, we developed a new metric called "CannyAvg" explicitly measuring seam visibility based on Canny edge detection. After testing, we numerically confirmed that each of our components significantly reduces seam visibility. We have added these quantitative ablation results to our revised manuscript.
>
> **Evaluation inconsistency** (addressing weakness 1)
>
> We apologize for the confusion regarding the CLIP evaluation metrics. In our revised manuscript, we have included example close-distance views sampled from the generated scene as well as more detailed descriptions about our evaluation method in Appendix A.3.1. We hope this strengthens the validity of our CLIP evaluation.
>
> **Limited controllability** (addressing weakness 2 and question 2)
>
> Sorry for the confusion. As you mentioned, Appendix A.1 gives an explicit example of the prompts for generating one scene. Because the prompts are arranged as a 3D tensor, prompts in spatial proximity in the 3D tensor will remain close in the generated scene. However, we do not force the user to provide a tensor that has the exact shape of our tiles, as it may limit the flexibility and ease of use. Instead, we treat the input prompt tensor as a tensor spanning the entire scene. Specifically, given a tile location, we sample the nearest prompt in the prompt tensor. This way, users can provide a coarser prompt tensor than the number of tiles.
>
> In case the user provides a finer prompt tensor, we have tried a strategy of blending prompts as demonstrated by PerpNeg: "Re-imagine the Negative Prompt Algorithm: Transform 2D Diffusion into 3D, alleviate Janus problem and Beyond” (https://arxiv.org/abs/2304.04968), but found limited improvements. Thus, to keep the simplicity of our pipeline, we did not include it in our method. We have clarified this in Appendix A.1 of our revised manuscript.
>
> ... continue ...

---

> ### Author Response · Authors · 2025-11-22
>
> **Low-resolution figures** (addressing weakness 3 and question 1)
>
> We apologize for the low-resolution figures in our initial submission. We have replaced those figures with the best quality images we can afford for the fixed upload size. We have also included several close-up views in Appendix A.3.1 that are consistent with our CLIP evaluation views.
>
> **Notation and Formatting**
>
> Thank you for your suggestions!
>
> (1) We have moved Figure 2 to page 4. $\mathcal{O}(\Delta t^2)$ denotes Big-O error resulting from Euler discretization. In practice, we do not include this term in our implementation as it is negligible compared to other sources of error. We have clarified this in the revised manuscript.
>
> (2) The subscript $\[\cdot\]_{f}$ $w_i^{(t)}(x, y, z)$denotes that, for each $w_i^{(t)}$, we perform Euler integration within the local tile space instead of the global scene space. We need to have this subscript because otherwise it would mean that the diffusion model $\theta$ has context of the entire scene.
>
> (3) $S$ (the size of diffusion context, or the size of a tile) is $16$ in practice. We do not have control over $S$ as it is determined by the architecture of the base diffusion model, as mentioned in Section 3.1. What was confusing to the reader might be the use of the words "tile" and "chunk". A clearer description is: A 4x3x1 chunk layout consists of 12 chunks, for $s = 16$, we get $5\times3=15$ tiles, and for $s = 8$, we get $7\times5=35$ tiles. (The exact tile number depends on the actual implementation, even for a fixed $s$ due to potential padding. We start our first tile in the middle and do not pad the edges. Whenever the tile exceeds the scene boundary, we shift it back to encourage more overlaps. Overlaps increase the quality of the generation as shown in Figure 8 of the revised manuscript.) The seemingly 8x6 layout is due to the fact that we set $s = \frac{S}{2} = 8$ for all experiments, including the autoregressive cases. We have revised our paper to clarify the distinction between “chunk” and “tile” in Section 3.2.
>
> (4) Thank you for your feedback. We have changed all related descriptions to "Average Blending."
>
> (5) We have added a `\paragraph` dedicated to tiled decoder in Section 3.3.
>
> > Thank you again for your constructive feedback! We have incorporated your suggestions and clarifications into our revised manuscript. Please let us know if you have any additional questions or concerns. We are happy to provide further clarifications or modifications as needed.

---

### Official Review · Reviewer_exxc · 2025-10-30

**Soundness:** 2
**Presentation:** 2
**Contribution:** 2
**Rating:** 2
**Confidence:** 4

**Summary:**

The paper aims to lift the ability of 3D object generation models to 3D world space. It introduces a training-free approach, trellisworld, to achieve 3D world generation. Based on a text-to-3D model, trellisworld denoises multiple 3D tiles in parallel and use an average weighting mechanism to aggregate the results at each timestep, thereby generating coherent 3D world.

**Strengths:**

Overall, the paper introduces a training-free method to generate 3d world using a 3D object generator. It provides a simple and effective method to achieve meaningfull applications.

**Weaknesses:**

The originality of the paper is somehow limited. The article claims the difference between them and syncity is that syncity depends on image inpainting, but this is merely a difference in the conditional mechanism. Aside from this conditional mechanism, the overall pipeline, which involves generating tiles and then blending, is very similar. Furthermore, the mechanisms of tile diffusion and blending mentioned in the paper are very similar to those of MultiDiffusion [1], and I haven't seen any effective strategies specifically designed for 3D.

[1] MultiDiffusion: Fusing Diffusion Paths for Controlled Image Generation

**Questions:**

1. The performance of TRELLIS in text-to-3D geneartion is worse than in image-to-3D generation. Therefore, I'm wondering if using text as the conditioned condition might have a lower performance, as I understand that generalization and controllability should be worse than models conditioned on images.
2. Quality of figures in the paper should be improved.
3. What is the maximum number of tiles that Trellisworld can generate simultaneously? How does the model's performance change as the number of simultaneously generated tiles increases?

---

> ### Author Response · Authors · 2025-11-22
>
> Dear Reviewer exxc,
>
> > Thank you for your recognition of our contributions to the simplicity and effectiveness of our method. We also appreciate your recognition of our contributions in advancing the field of text-to-3D scene generation.
>
> **Limited Originality** (addressing weakness 1)
>
> SynCity and our approach are completely different implementations tackling the same problem. Indeed, SynCity is the only other method we know of that tackles the problem of converting an object generator to a scene generator without the need of an LLM. We both chose to leverage TRELLIS as the base object generator for the quality of TRELLIS and fairness in our comparison, and the use of the base model is the only similarity between our methods.
>
> We agree that our method involves blending inside diffusion steps, which is similar to MultiDiffusion in the image domain. One of the goals of our work is to demonstrate that such a blending strategy can be effectively applied to 3D generation as well. Doing so unlocks the scene-level generation capability of the base model (which is different from increasing image size shown in MultiDiffusion). The publication of our paper would potentially inspire more research towards leveraging existing 3D object generators for cohesive scene generation. This is important because as the research community lacks large-scale open-vocabulary scene datasets, our idea of leveraging object generators can be crucial for the development of future 3D scene generation methods.
>
> We have made adaptations to 3D-specific challenges as well.
> (1) Elevation to 3D is not trivial: It is not safe to say that blending in 3D work by providing only 2D evidence given by MultiDiffusion. The following thought experiment shows why elevation to 3D is not trivial: Consider a diffusion model $\theta: \mathbb{R}^2 \to \mathbb{R}^2$ trained only on an image consists of 2 grey-scale pixels $(0, 255)$. When we try to tile this diffusion to generate 3 pixels, we might observe that the middle pixel won’t converge: $(0, 123, 255)$. This example shows that the success of blending depends on the distribution we try to model. It is unsafe to assume that the latent 3D distribution shares the same structure as the latent 2D distribution because the information density for 3D is believed to be sparser than 2D (this is the underlying assumption of many 3D compression algorithms such as Triplane). If the 3D latent space is anisotropic (which can happen if the base model VAE is trained with a larger compression rate with respect to the 3D information density), then such blending may not be possible. We are fortunate enough to find out it is not the case.
> (2) Blending with sparse tensor: Images in 2D are often represented with latent dense tensors whereas 3D latent representations can use sparse tensor. To our knowledge, we are the first to show that blending in sparse denoising spaces (just like TRELLIS SLAT) can still be effective.
> (3) Tiling the decoder: Different from image diffusion though, we found out that tiling the 3D decoder is strictly necessary for generating 3DGS, as shown in ablation Figure 5.
>
> A more 3D-specific method (such as encouraging the connectivity between different chunks) could be implemented. However, doing so not only adds complexity to our framework but also makes it impossible to generate “Group of Fish (2x2x2 chunk)” and “Castle in the Sky (1x1x2 chunk)” in our figure. This ability is a clear advantage of our method compared to SynCity (a 3D-specific method).
>
> In addition, we extended MultiDiffusion by having:
> (4) A more effective blending strategy: MultiDiffusion blends with binary masks, which correspond to the average blending method in our Table 1. We showed that the cosine blending strategy produces more cohesive results compared to average blending.
> (5) New metric: In our revised manuscript, we have included a metric to quantify seams in Section 4.3.
>
> ... continue ...

---

> ### Author Response · Authors · 2025-11-22
>
> **TRELLIS in text-to-3D generation** (addressing question 1)
>
> As mentioned by TRELLIS authors, "It is always recommended to do text-to-3D generation by first generating images using text-to-image models and then using TRELLIS-image models for 3D generation. Text-conditioned models are less creative and detailed due to data limitations." However, using image-conditioned models (as demonstrated by SynCity) will result in less coherent blending as well as significantly higher computational cost due to complexity in their pipeline. For scene generation, our contribution demonstrated that the text-conditioned approach is better than the image-conditioned approach both in quality and efficiency. This lowering of performance might be true for single-object generation, but for scene generation, we believe the text-conditioned approach is better and therefore should be advocated to the 3D generation community.
>
> **Figure quality** (addressing question 2)
>
> We apologize for the low-resolution figures and non-SVG figures in our initial submission. We have replaced those figures with the best quality images we can afford for the fixed upload size.
>
> **Maximum number of tiles** (addressing question 3)
>
> The largest number of tiles we have tried is 672 tiles (on a 16GB GPU). We have not yet reached the limit of our method but we stopped simply due to computational time. We have additionally included relevant runtime analysis in Figure 8, Figure 9, and Figure 10. As shown in these figures, the computational time scales linearly with respect to the number of tiles. We believe our method can scale to even larger scenes as long as sufficient computational resources are provided. We do not observe an increase of memory usage for larger tiles.
>
> > Thank you for your critical feedback as it helped us improve our manuscript. We have incorporated your suggestions and clarifications into our revised manuscript. Please let us know if you have any additional questions or concerns. We are happy to provide further clarifications or modifications as needed.

---

> > ### Comment · Reviewer_exxc · 2025-11-26
> >
> > I somehow agree that this work has some incremental technical innovation, so I would consider raising the score to 4. I think this is the highest score I can give. I still have two concerns:
> >
> > I don't agree that applying multidiffusion to 3D would be too difficult. This is because, compared to vecsets, the voxel representation used by Trellis is very similar in nature to image representation. Furthermore, regarding text versus image conditions, is there more sufficient experimental evidence to show that using text as a condition is superior to image conditions in your current approach?

---

> > > ### Author Response · Authors · 2025-12-04
> > >
> > > Dear Reviewer exxc,
> > >
> > > **voxel representation**
> > >
> > > Thank you for recognizing our work. You are right that TRELLIS’s voxel representation is more similar to pixels than to vector sets. This is the main reason we chose to experiment with TRELLIS.
> > >
> > > **text versus image conditions**
> > >
> > > You brought up a good point about text versus image conditions, and we think sharing our experimental and theoretical insights may be beneficial to the community. We have added a detailed discussion with additional experimental results to our paper in Appendix A.5 to justify why text-conditioned diffusion is better at blending than image-conditioned diffusion. Put simply, we suspect this performance discrepancy arises because the learned conditional distribution associated with the text score model is more diffuse than that of the image model.

---

### Official Review · Reviewer_9vZM · 2025-10-31

**Soundness:** 3
**Presentation:** 3
**Contribution:** 3
**Rating:** 6
**Confidence:** 2

**Summary:**

The authors propose TRELLISWorld, a training-free framework for text-driven 3D scene generation, which composes complex scenes by leveraging pre-trained text-to-3D object diffusion models.
Instead of training an end-to-end scene generator, TRELLISWorld decomposes the scene noise into multiple object-level subregions (“chunks”) and employs a cosine-weighted re-aggregation strategy to efficiently synthesize large-scale 3D environments.
Compared to the state-of-the-art method SynCity, TRELLISWorld achieves superior visual quality and significantly faster inference, demonstrating the effectiveness of its modular and scalable generation approach.

**Strengths:**

1. State-of-the-Art Results
The proposed TRELLISWorld achieves superior CLIP score performance compared to the recent state-of-the-art method SynCity, while also requiring less computational resources and delivering faster inference speed. This demonstrates the efficiency and scalability of the training-free design.

2. Comprehensive Ablation Studies
The authors present comprehensive qualitative ablation studies on key components—Tiled Diffusion, Blending, and Tiled Decoder—clearly illustrating the contribution of each to the final scene generation quality. These studies effectively highlight how each module enhances visual coherence and overall realism.

**Weaknesses:**

1. Heavy Reliance on the Base Model
As acknowledged in the manuscript, the proposed method—being training-free—is inherently limited by the capabilities of its underlying base model, TRELLIS. Consequently, the overall performance and generalization ability are closely tied to the pretrained model’s strengths and weaknesses, which may restrict the method’s applicability across diverse domains.

2. Lack of Quantitative Ablation Studies
While the qualitative ablation studies provide valuable insights, the paper would benefit from quantitative analyses to numerically assess the contribution of each component. Such evaluations would help clarify how elements like Tiled Diffusion, Blending, and Tiled Decoder quantitatively influence the final output quality and performance.

**Questions:**

1. Could the authors investigate how the performance changes when the base model is replaced with alternatives to TRELLIS? Such an analysis would help assess the generality and adaptability of the proposed framework.

2. It is assumed that the stride size used in the tiled generation process may influence the final performance. Could the authors conduct additional experiments with varying stride sizes to analyze its impact on scene quality and consistency?

3. Could the authors provide quantitative ablation results (e.g., CLIP score) without the proposed components to clarify the contribution of each module to the overall performance?

---

> ### Author Response · Authors · 2025-11-22
>
> Dear Reviewer 9vZM,
>
> > Thank you for your time and valuable feedback on our submission. As you pointed out, we have achieved both superior quantitative results and require less computational resources compared to the recent state-of-the-art method. Our qualitative evaluation justified each component of our method, keeping each of our modules simple and effective.
>
> **Reliance on base-generator** (addressing weakness 1)
>
> Indeed, our method's performance is influenced by the capabilities of the underlying text-to-3D object generator. This trait is common for inference methods that build upon pre-trained models. This means that future advancements in base generators directly translate to improved scene generation quality in our framework because our method is an add-on improvement of the base generators. Furthermore, we believe our simple approach provides a flexible framework that can be adapted to any text-based 3D diffusion object generator with minimal modifications according to their model architecture.
>
> **Lack of quantitative ablation** (addressing weakness 2 and question 3)
>
> Our submission does include quantitative ablation studies on the components of our method in Table 1. We have evaluated the impact of (1) tiled diffusion,  and (2) weighted blending on CLIP score. Removing any of the components results in a decrease in performance, justifying their necessity in our final method. To further strengthen our ablation studies, we have added additional experiments to explicitly quantify the seam visibility for each ablation setting using the Canny edge-based seam metric described in our evaluation section. The results confirm that both tiled diffusion and weighted blending contribute to reducing seams in the generated scenes.
>
> **Extension to other base generators** (addressing question 1)
>
> Thank you for your suggestions. We have added performance analysis for base generators other than TRELLIS. To this end, we have included qualitative results of applying our method to two different 3D diffusion generators with different architectures and trained domains. While both are only trained on 16x16x16 voxels, they are able to generate larger terrains or structures while keeping the generation cohesive. Qualitative results can be found in Appendix A.4 of our revised manuscript.
>
> **Scene quality by stride size** (addressing question 2)
>
> Thank you for your suggestion. We have conducted additional experiments to evaluate the impact of stride size on scene quality as well as computational cost. We varied the stride size and measured seam visibility using our new metric. The results indicate that smaller stride sizes lead to reduced seam visibility, confirming that finer overlaps between tiles enhance overall scene coherence. We have also justified the optimal stride value we chose to achieve the optimal trade-off between computational cost and seam visibility. We have included these results in Table 1.
>
> > Thank you for providing feedback that helped us improve our manuscript. We have incorporated the suggested changes and clarifications. Please let us know if you have any additional questions or concerns. We are happy to provide further clarifications or modifications as needed.

---

### Official Review · Reviewer_RN1P · 2025-10-31

**Soundness:** 3
**Presentation:** 3
**Contribution:** 3
**Rating:** 4
**Confidence:** 3

**Summary:**

The paper targets the training-free goal for 3d scene generation. The authors propose TRELLISWorld, which reframes text-to-3D scene synthesis as a multi-tile denoising problem. The overlapping 3D regions are generated by a pretrained object-level model and blended with cosine-weighted averaging. The experiments show qualitative results with advantages over SynCity.

**Strengths:**

The core idea of using tiled diffusion with cosine blending to smoothen the inter-tile transition is straightforward with easy-to-understand intuition.

The method description is clear and the implementation provides some details, though it's doubtful if it's sufficient for readers to reperform w/o open-sourced codes.

The results show clear advantages over the peering work Syncity.

The limitation section acknowledges its base-model dependence and lack of object disentanglement.

**Weaknesses:**

As mentioned in the strength, the method is quite straightforward, therefore the impact heavily lies in the provision of the tool as opensourced code to the community, as SynCity has done.

The innovative contribution is more an incremental improvement of Trellis, thus whether it meets the standard as a standalone paper in ICLR may need further discussion.

The work is heavily depending on the base object generator, which limits the contribution.

The comparison is mainly against SynCity while other recent 3Descene generation works referenced in the related work sections are largely missing.

The computation cost analysis is too simple, without showing any memory/runtime tests scaling with tile counts or comparisons to optimized SDS/LRM pipelines.

**Questions:**

1. Could the authors provide complete implemenation details with full metric setups?
2. Could the authors broaden the tests addressing the comments in the weakness, e.g. scale up with tile count to test computation, add other 3d scene baselines etc.?
3. Does the proposal only work for static scene? Any idea how to make it work on dynamic scene?
4. Can this proposal function as well on other base generator besides Trellis? Could the authors test the performance impact across a few other base generators?

---

> ### Author Response · Authors · 2025-11-22
>
> Dear Reviewer RN1P,
>
> > Thank you for your constructive feedback. We appreciate your positive comments regarding the simplicity of our approach and its intuition, as trimming down unnecessary complexity is indeed one of our main goals and advantages compared to SynCity. Doing so not only increased result generation quality but also significantly cut down the computational resources needed.
>
> **Opensourcing the model** (addressing weakness 1 and question 1)
>
> We have added a statement in our abstract indicating that we will open-source both our code and evaluation methods for community impact upon acceptance of the paper. We are committed to making our work accessible to researchers and practitioners in the field.
>
> **Incremental improvements over TRELLIS** (addressing weakness 2)
>
> We acknowledge that our current work builds upon the TRELLIS framework. However, we believe that our contributions, particularly in blending text-based object generators for scene generation, point out a new research direction for the 3D generation research community. The publication of this paper would potentially inspire more research towards leveraging existing 3D object generators for cohesive scene generation. This is important because as the research community lacks large-scale open-vocabulary scene datasets, our idea of leveraging object generators can be crucial for the development of future 3D scene generation methods.
>
> **Dependency on base-generator** (addressing weakness 3)
>
> Indeed, our method's performance is influenced by the capabilities of the underlying text-to-3D object generator. This trait is common for inference methods that build upon pre-trained models. This means that future advancements in base generators directly translate to improved scene generation quality in our framework because our method is an add-on improvement to the base generators. Furthermore, we believe our simple approach provides a flexible framework that can be adapted to any text-based 3D diffusion object generator with minimal modifications according to their model architecture.
>
> **Missing comparisons to works mentioned in Related Works** (addressing weakness 4)
>
> As mentioned, we do not show comparisons to (1) 3D reconstruction methods (2) scene generators based on 2D generation (3) 3D native generation (4) LLM-based methods. We believe it is difficult to establish an unbiased comparison to these works for the following reasons:
> (1) 3D reconstruction methods require multi-view images as input, which is a different task setting compared to our task which generates 3D scenes from text prompts alone.
> (2) Scene generators based on 2D generation often do not produce a 360-degree viewable 3D representation, making it challenging to evaluate them on the same metrics as our method since we calculated CLIP using images sampled from 360-degree rotation and evaluated seams based on orthographic projections. Furthermore, we cannot compare computational efficiency under the same generated resolution because 2D-video generated 3DGS does not have uniform resolution across the 3D space.
> (3) As mentioned in Related Works, current 3D native generation methods are limited to a closed set of object classes which makes it hard to establish a fair comparison on open-vocabulary scene generation tasks.
> (4) LLM-based methods generate individual objects floating mid-air without any scene context. They do not try to blend between objects or generate ground or background elements, making it both difficult and less meaningful to compare the scene quality.
>
> For the above reasons, we focused our comparisons on methods that share the same task setting. We hope this resolves your concerns.
>
> **Cost analysis is too simple** (addressing weakness 5 and question 2)
>
> We appreciate your feedback! We have added a more detailed analysis of the computational cost, including per-tile processing time and its scalability with respect to the number of tiles and stride size. We have also included a graph showing the optimal trade-off between computational cost and seam visibility in our revised manuscript. We focus our comparison with SynCity as we can ensure we are compared on the same resolution and diffusion steps. If we were to compare to object- or single-room-scale generators, our method, in the general sense, is faster than typical SDS-based methods (as they involve more denoising steps and test-time optimization), but slower than LRM-based methods (as they do not denoise). However, LRM-based methods typically do not generate beyond object scale.
>
> ... continue ...

---

> ### Author Response · Authors · 2025-11-22
>
> **Dynamic Scene Generation** (addressing question 3)
>
> Thank you for your insightful comments. Dynamic scene generation is outside the scope of this work, however, it is indeed an interesting direction to explore. One straightforward approach may be blending the static scene and then generating the animations separately, which also provides more control to the users. Blending between objects and animations simultaneously may be possible and here are our preliminary thoughts: One approach would be to apply the same method described in our paper. However, this is unlikely to achieve meaningful results because our method leverages the almost one-to-one correspondence between the generated 3D location and the location of the diffusing latent representation. Since time-based diffusion models generally have a more efficient latent representation than a full 4D representation, the same approach needs to be tuned on the specific model architecture for it to work. We are happy to explore this direction in future works.
>
> **Extension to other base generators** (addressing question 4)
>
> The framework does not rely on architectural details unique to TRELLIS. Instead, it assumes only a diffusion-based text-to-3D object generator. In theory, our method transfers to other object-level 3D diffusion models. To demonstrate, we have included qualitative results of applying our method to two different 3D diffusion generators with different architectures and trained domains. While both are only trained on 16x16x16 voxels, they are able to generate larger terrains or structures while keeping the generation cohesive. Qualitative results can be found in Appendix A.4. Qualitative results can be found in Appendix A.4 of our revised manuscript.
>
> > Once again, thank you for your insightful review. We have incorporated the suggested changes into our paper. Please let us know if you have any additional questions or concerns. We are happy to provide further clarifications or modifications as needed.

---

### Author Response · Authors · 2025-12-04

Dear AC,

We would like to thank all reviewers and the AC for their time and valuable feedback, which greatly helped us further improve our manuscript.

**Paper Summary**: Our work introduces a training-free method for text-to-3D scene generation that repurposes general text-to-3D object diffusion models as modular tile generators. Our method eliminates the need for scene-level datasets or retraining, speeds up computation, and surpasses existing methods in perceptual alignment and seam visibility both quantitatively and qualitatively.

**Rebuttal Summary**:

We have addressed all weaknesses and questions from all reviewers. In particular, we:
- clarified that we will open-source our method (RN1P)
- clarified our contribution over TRELLIS and Syncity (RN1P, exxc)
- added experiments with other base-generators (RN1P, 9vZM)
- clarified why we chose not to include irrelevant comparisons (RN1P)
- added detailed cost and scalability analysis (how stride size and number of tiles affect quality and efficiency) to our manuscript (RN1P, 9vZM, exxc)
- clarified why dynamic scene generation is out of scope for this manuscript (RN1P)
- strengthened our ablation by adding a new metric (9vZM)
- clarified the differences between us and Syncity (exxc)
- clarified our adaptation to 3D-specific challenges and improvements to MultiDiffusion in 2D (exxc)
- improved figure quality (exxc, 7XGV)
- clarified misunderstandings about speed improvements (7XGV)
- added details about our evaluation method (7XGV)
- clarified and/or fixed symbols and formatting (7XGV)

Due to the special circumstances this year, only Reviewer exxc was able to participate in the discussion before Nov 26. Please refer to the discussion below.

Sincerely,
TRELLISWorld Authors

---

### Meta-Review · Area_Chair_JcwD · 2026-01-06

**Summary:**

The paper present a method for generating 3D worlds using pretrainined text-to-3D generator. The method is based on combination of two methods Multidiffusion and Trellis, and is essentially Multidiffusion in 3D domain.

Two main concerns left unaddressed, according to area char, are incremental contribution (RN1P and exxc) and reliance on text-to-3D version of 3D generator (exxc and 7XGV), which is in general weaker than image-to-3D.

Given these issues area chair recommends a rejection.

**Reviewer Concerns:**

*RN1P* -

**opensource**, **dependence on base generator**, *cost analysis* - these issues was addressed according to area chair.

**incremental improvement** - this issue was not addressed and was also mentioned by exxc.

**missing comparisons with the baselines from related works** - this issue was not addressed, authors could provide qualitative evaluations or user studies if quantitative evaluation is not possible.

*9vZM* -

**dependence on the base generator**, **lack of Quantitative Ablation Studies** - these issues was addressed according to area chair.

*exxc* -

**originality** - this issue was not addressed and was also mentioned by RN1P.

**inability to utilize image-to-3D mode of the base generator** - this issue was also not addressed and also mentioned by 7XGV.

*7XGV* -

**evaluation inconsistency** and **figure resolution** - these issues was addressed according to area chair.

**limited controlability, inability to utilize image-to-3D** -  this issue was also not addressed and also mentioned by exxc.

**Reviewer Scores:**

RN1P - stay the same

9vZM - stay the same

exxc - increase to 4

7XGV - stay the same

---

### Decision · Program_Chairs · 2026-01-26

Reject